# `cfr` (v2024.1.26): a Python package for climate field reconstruction

Feng Zhu[1], Julien Emile-Geay[2], Gregory J. Hakim[3], Dominique Guillot[4], Deborah Khider[5], Robert Tardif[3], and Walter A. Perkins[6]

[1]Climate and Global Dynamics Laboratory, National Center for Atmospheric Research, Boulder, CO, USA
[2]Department of Earth Sciences, University of Southern California, Los Angeles, CA USA
[3]Department of Atmospheric Sciences, University of Washington, Seattle, WA USA
[5]University of Southern California, Information Sciences Institute, Marina Del Rey, CA
[4]Department of Mathematical Sciences, University of Delaware, Newark, DE, USA
[6]Allen Institute for Artificial Intelligence, Seattle, WA USA

**Correspondence:** Feng Zhu (fengzhu@ucar.edu)

**Abstract.** Climate field reconstruction (CFR) refers to the estimation of spatiotemporal climate fields (such as surface temperature) from a collection of pointwise paleoclimate proxy datasets. Such reconstructions can provide rich information on climate dynamics and provide an out-of-sample validation of climate models. However, most CFR workflows are complex and time-consuming, as they involve: (i) preprocessing of the proxy records, climate model simulations, and instrumental observations, (ii) application of one or more statistical methods, and (iii) analysis and visualization of the reconstruction results. Historically, this process has lacked transparency and accessibility, limiting reproducibility and experimentation by non-specialists. This article presents an open-source and object-oriented Python package called `cfr` that aims to make CFR workflows easy to understand and conduct, saving climatologists from technical details and facilitating efficient and reproducible research. `cfr` provides user-friendly utilities for common CFR tasks such as proxy and climate data analysis and visualization, proxy system modeling, and modularized workflows for multiple reconstruction methods, enabling methodological intercomparisons within the same framework. The package is supported with an extensive documentation of the application programming interface (API) and a growing number of tutorial notebooks illustrating its usage. As an example, we present two `cfr`-driven reconstruction experiments using the PAGES 2k temperature database: applying the last millennium reanalysis (LMR) paleoclimate data assimilation (PDA) framework and the Graphical Expectation-Maximization (GraphEM) algorithm, respectively.

## 1 Introduction

Paleoclimate reconstructions provide critical context for recent changes and out-of-sample validation of climate models (Masson-Delmotte et al., 2013). Site-based reconstructions have been used to infer climate change with the assumption that site-based paleoclimate records (i.e., proxies) can represent the regional or even global climate variations to some extent (e.g., Shakun et al., 2012; Marcott et al., 2013), yet such approaches may lead to conclusions biased to specific sites (Bova et al., 2021; Osman et al., 2021). Climate field reconstruction (CFR), in contrast, aims to optimally combine the available proxy data and infer multivariate climate variability over an entire domain of interest, hence alleviating the biases in site-based reconstruction methods. It has become the emerging approach to studying spatiotemporal climate history and model-data comparisons over

the past 2,000 years (e.g. Tingley et al., 2012; Neukom et al., 2019a; Tierney et al., 2020; Zhu et al., 2020; Osman et al., 2021; King et al., 2021; Zhu et al., 2022).

Existing CFR methods can be classified into three main categories. The first category is based on regression models, which fit climate variables (e.g., temperature, pressure, precipitation, and so on) onto proxy observations using some variant of least squares (e.g., Mann et al., 1998, 1999; Evans et al., 2002; Cook et al., 2004, 2010; Tingley et al., 2012; Smerdon and Pollack, 2016). A more formal approach is based on Bayesian hierarchical models, which explicitly model the various levels of the relationship linking climate processes to proxy observations, and invert the proxy-climate relation via Bayes' theorem (e.g.,

Tingley and Huybers, 2010a, b, 2013; Christiansen and Ljungqvist, 2017; Shi et al., 2017). Both approaches rely on the covariance structure between proxies and climate variables that is estimated from observational data over a calibration period. The third is based on paleoclimate data assimilation (PDA, e.g., Dirren and Hakim, 2005; Goosse et al., 2006b; Ridgwell et al., 2007; Steiger and Hakim, 2016; Hakim et al., 2016; Tardif et al., 2019), which follows a similar idea to the previous approaches, but with a critical difference that the proxy-climate covariance matrix is estimated from climate model simulations

instead of observational data, as such spatial, multivariate relationships are available and subject to dynamical constraints. As a result, PDA methods can, in principle, estimate any field simulated in the model prior, though the reconstruction quality will be a function of how strongly those variables can be constrained by paleo observations.

Since some CFR methods have been shown to depend sensitively on the input data (Wang et al., 2015), it is imperative to apply more than one method to the same problem to establish a result's robustness. This has heretofore been difficult, as

most CFR workflows are complex, possibly involving the selection and processing of proxy records, the processing of gridded climate model simulation and instrumental observation data, the calibration of proxy system models (PSMs, Evans et al., 2013), the setup and execution of the specific reconstruction algorithms, as well as the validation and visualization of the reconstruction results. Each of these steps can lead to a long decision tree (Bürger et al., 2006; Büntgen et al., 2021), complicating comparisons. Furthermore, these steps require a comprehensive knowledge of proxies, data analysis and modeling, reconstruction methods, as well as scientific programming and visualization, and can be obscure and error-prone for climatolo-

gists not familiar with all of them. Here we present a Python package called `cfr` that is designed to make CFR workflows easy to understand and perform, with the aim of improving the efficiency and reproducibility of CFR-related research.

The paper is organized as follows: Sect. 2 presents the design philosophy of the package. Sect. 3 provides a primer for the two classes of reconstruction methods presently included in the package. Sect. 4 introduces the architecture and major

modules. Sect. 5 and 6 describe the reconstruction workflows of two methods: paleoclimate data assimilation and graphical expectation-maximization, respectively. A summary and an outlook are provided in Sect. 7.

## 2   The `cfr` design philosophy

`cfr` aims to provide intuitive and effective workflows for climate field reconstruction with the following features:

**Reproducible Computational Narratives.** Although a traditional script-based style is also supported, `cfr` is natively de-
signed to be used in the context of reproducible computational narratives known as Jupyter notebooks (Kluyver et al., 2016),

which provides an interactive programming laboratory for data analysis and visualization, and has become the new standard for analysis-driven scientific research.

**Intuitive.** `cfr` is coded in the object-oriented programming (OOP) style, which provides intuitive ways to manipulate data objects. For instance, each proxy record object (`ProxyRecord`) supports a collection of methods for analysis and visualization; proxy record objects can be added together (by the operator "+") to form a proxy database object (`ProxyDatabase`) that supports another collection of methods specific to a proxy database. So-called "method cascading" is also supported to allow for smooth processing of the data objects in one line of code combining multiple processing steps.

**Flexible.** `cfr` provides multiple levels of modularization. It supports object-specific workflows, which provide in-depth operations of the data objects, as well as workflows specific to reconstruction tasks, enabling macroscopic manipulations of the reconstruction-related tasks.

**Community-based.** `cfr` is built upon community efforts on (paleoclimate) data analysis, modeling, and visualization, including but not limited to NumPy (van der Walt et al., 2011), SciPy (Virtanen et al., 2020), Pandas (McKinney, 2010), Xarray (Hoyer and Hamman, 2017), Matplotlib (Hunter, 2007), Cartopy (Met Office, 2010), Seaborn (Waskom, 2021), Plotly (Plotly Technologies Inc., 2015), Statsmodels (Seabold and Perktold, 2010), Pyleoclim (Khider et al., 2022), and PRYSM (Dee et al., 2015). This makes the codebase of `cfr` not only concise and efficient, but also ready for grassroots open development.

**User-friendly.** `cfr` is designed to be easy to install and use. It is supported with an extensive documentation on the installation and the essential application programming interface (API), as well as a growing number of tutorial notebooks illustrating its usage. In addition, the commonly used proxy databases and gridded climate data for CFR applications can be remotely loaded from the cloud with `cfr`'s data fetching functionality.

## 3 Reconstruction Methods

At the moment, `cfr` supports two classes of reconstruction methods, though it is designed to accommodate many more.

### 3.1 Offline Paleoclimate Data Assimilation (PDA)

In recent years, paleoclimate data assimilation (PDA) has provided a novel way to reconstruct past climate variations (Jones and Widmann, 2004; Goosse et al., 2006a; Gebhardt et al., 2008; Widmann et al., 2010; Goosse et al., 2010; Annan and Hargreaves, 2013; Steiger et al., 2014; Hakim et al., 2016; Franke et al., 2017; Acevedo et al., 2017; Steiger et al., 2018; Shi et al., 2019; Tierney et al., 2020; Osman et al., 2021; King et al., 2021; Lyu et al., 2021; Zhu et al., 2022; Valler et al., 2022; Annan et al., 2022; Fang et al., 2022). PDA proceeds by drawing from a prior distribution of climate states, which it updates by comparison with observations (Wikle and Berliner, 2007). The prior comes from physics-based model simulations, typically from general circulation models experiments covering past intervals (e.g. Otto-Bliesner et al., 2016).

The observations, in this case, are the values of climate proxies, which indirectly sense the climate field(s) of interest (e.g. surface temperature, precipitation, wind speed). The link between latent climate states and paleo proxy observations is provided by observation operators (one for each proxy type), which in this case are called proxy system models (PSMs, Evans

et al., 2013). PSMs are low-order representations of the processes that translate climate conditions to the physical or chemical observations made on various proxy archives, such as tree rings, corals, ice cores, lake and marine sediments, or cave deposits (speleothems).

`cfr` implements a version of a data assimilation algorithm known as an offline ensemble Kalman filter (EnKF), popularized in the paleoclimate context by Steiger et al. (2013); Hakim et al. (2016); Tardif et al. (2019). The code for the EnKF solver is derived from the Last Millennium Reanalysis (LMR) codebase (https://github.com/modons/LMR) and was streamlined with utilities from the `cfr` package. The output of this algorithm is a time-evolving distribution (the "posterior") quantifying the probability of particular climate states over time.

PDA can reconstruct as many climate fields as are present in the prior, though they will not be equally well constrained by the observations. Prior values are usually obtained by drawing $n$ (100 to 200) samples at random from a model simulation at the outset. This precludes sampling variations, ensuring that all variability in the posterior is driven by the observations.

Uncertainty quantification is carried out by random sampling of a subset of the proxy set (typically 75%) for separate training and validation. This process is typically repeated MC (20-50) times, yielding so many Monte-Carlo "iterations" of the reconstruction, each of which contains a unique set of $n$ samples from the prior. The posterior is therefore composed of contains $n \times$ MC trajectories, typically numbering in the thousands. For scalar indices like the global mean surface temperature or the North Atlantic Oscillation index, the entire ensemble is exported. However, for climate fields, this added dimension would yield unacceptably large files. As a compromise, for each field variable, `cfr` by default exports the ensemble mean of the $n$ reconstructions during each Monte-Carlo iteration, and the final output thus contains MC ensemble members. In case the full ensemble is needed, `cfr` also provides a switch to export the entire $n \times$ MC members of the reconstructed fields.

The framework treats each target year for reconstruction in a standalone manner, i.e. the reconstruction of a certain year will not affect that of the others. This design is a key feature of offline DA. It alleviates the issue of non-uniform temporal data availability and facilitates the parallel reconstruction of multiple years.

## 3.2 Graphical Expectation Maximization (GraphEM)

GraphEM (Guillot et al., 2015, hereafter G15) is a variant of the popular Regularized Expectation-Maximization (RegEM) algorithm introduced by Schneider (2001), and popularized in the paleoclimate reconstruction context by Michael Mann and co-authors (Rutherford et al., 2005; Mann et al., 2007; Mann et al., 2008, 2009; Emile-Geay et al., 2013a, b). Like RegEM, GraphEM mitigates the problems posed to the traditional EM algorithm (Dempster et al., 1977) by the presence of rank-deficient covariance matrices, due the "large $p$, small $n$" problem: climate fields are observed over large grids (e.g. a $5° \times 5°$ global grid contains nearly 2500 grid points) but a comparatively short time (say 170 annual samples for the period 1851-2020). GraphEM addresses this issue by exploiting the conditional independence relations inherent to a climate field (Vaccaro et al., 2021): namely, surface temperature at a given gridpoint is conditionally independent of the vast majority of the rest of the grid, except for a handful of neighbors, typically a fraction of a percent of the total grid size. Effectively, this reduces the dimensionality of the problem, to the point that the estimation is well-posed, and the various submatrices of the covariance matrix $\Sigma$ are invertible and well estimated.

The degree of regularization is determined by the density of non-zero entries in the inverse covariance matrix $\Omega$, which can be used to construct a graph adjacency matrix (Lauritzen, 1996). The more zero entries in this matrix, the sparser the graph, and the more damped the estimation. The fuller the graph, the more connections are captured between distant locales, but the more potential there is for a non-invertible covariance matrix, resulting in unphysical values of the reconstructed field. Therefore, the quality of the GraphEM estimation hinges on an appropriate choice of graph, which is a compromise between the two poles described above. In practice, it is helpful to split $\Omega$ into three parts:

$\Omega_{FF}$  stands for "field-field", and encodes covariances intrinsic to a given field, for instance covariance between temperature at different grid points.

$\Omega_{FP}$  stands for "field-proxy", and encodes the way the proxies covary with the target field (here, temperature).

$\Omega_{PP}$  stands for "proxy-proxy", and encodes how the values of one proxy may be predictive of another.

This last part is the identity matrix, because a proxy cannot provide any more information about another proxy than the underlying climate field can (Guillot et al., 2015). Therefore, the only parts of $\Omega$ that need be specified are $\Omega_{FF}$ and $\Omega_{FP}$. `cfr` supports two ways to do so:

**Neighborhood graphs** are the simplest method, identifying as neighbors all gridpoints that lie within a certain radius (say, 1000 km) of a given location. The smaller the radius, the sparser the graph; the larger the radius, the fuller the graph. Neighborhood graphs take no time to compute, but their structure is fairly rigid, as its sole dependence on distance means that it cannot exploit anisotropic features like land/ocean boundaries, orography, or teleconnection patterns.

**Graphical LASSO** (GLASSO) is an $\ell_1$ penalized likelihood method introduced by Friedman et al. (2008) in the context of high-dimensional covariance estimation. A major advantage of GLASSO is that it can extract non-isotropic dependencies from a climate field (Vaccaro et al., 2021). In `cfr`, the level of regularization is controlled by two sparsity parameters, which explicitly target the proportion of non-zero entries in $\Omega$ (see G15 for details): `sp_FF` controls the sparsity of submatrix $\Omega_{FF}$, and `sp_FP` controls the sparsity of submatrix $\Omega_{FP}$. The smaller the sparsity parameters, the sparser the graph, and the more damped the estimate.

G15 found that for suitable choices of the sparsity parameters, the GLASSO approach yielded much better estimates of the temperature field than the neighborhood graph method. However, the GLASSO approach has two main challenges: (1) the graph optimization can be computationally intensive and (2) it requires a complete data matrix to operate. In the case of paleoclimate reconstructions, this means that proxy series and instrumental observations of the climate field of interest must not contain any missing values over the entire calibration period (e.g. 1850-2000).

To address the first challenge, `cfr` uses a greedy algorithm to find the optimal `sp_FP` and `sp_FF`, as proposed by G15. To address the second challenge, we use GraphEM with neighborhood graphs to obtain a first guess for the climate field, use it to run GLASSO and obtain a more flexible graph, which is then used within GraphEM on the original (incomplete) data matrix (Section 6). Another advantage of this method is that it provides GraphEM (an iterative method) with a reasonable first

guess for the parameters of the distribution $(\mu_0, \Sigma_0)$, which can considerably speed up convergence. We call this approach the
"hybrid" method in `cfr`.

## 3.3 Generalization

The `cfr` framework is designed to be quite general, and can in principle accommodate any CFR method. PDA and GraphEM are only two of the methods used, for instance, in Neukom et al. (2019b), which are themselves a subset of all possible CFR methods. The two options included at this stage, based on completely different assumptions and methodologies, provide a way to test the method-dependence of a given result. This is critical, as some CFR methods have been shown to depend sensitively on the input data (Wang et al., 2015). More methods will be added into the package in subsequent versions. We also welcome of contributions from the community for the inclusion of more other Python-based CFR methods. A contributing guide can be accessed at: https://fzhu2e.github.io/cfr (Zhu et al., 2024).

## 4 The `cfr` architecture

To make CFR workflows intuitive and flexible, `cfr` takes an object-oriented programming (OOP) approach and defines Python classes describing and organizing the essential data objects along with their connections, such as proxy records, proxy databases, gridded climate fields, as well as the PSMs and reconstruction method solvers, under different modules, which are described in the following sections.

### 4.1 `proxy`: proxy data processing and visualization

The `proxy` module contains a `ProxyRecord` class and a `ProxyDatabase` class, providing a collection of operation methods for proxy data processing and visualization, which is fundamental for paleoclimate data analysis.

The `ProxyRecord` class provides a structure to store, process, and visualize a proxy record. The essential attributes and methods are listed in Table 1. For instance, suppose we have a `ProxyRecord` object named `pobj`, then calling `pobj.plot()` will visualize the proxy time series along with its location depicted on a map (Fig. 1). Basic metadata stored as its attributes will be displayed on the plot, including the proxy ID, the location in latitude and longitude, the proxy type, and the proxy variable with the unit, if available. There are methods for basic processing: calling `pobj.slice()` will slice the record based on the given timespan (while "index slicing" is also supported, please refer to the notebook tutorial: https://fzhu2e.github.io/cfr/notebooks/proxy-ops.html#Slice-a-ProxyRecord (Zhu et al., 2024) for the details), calling `pobj.annualize()` will annualize/seasonalize the record based on the specified list of months, and calling `pobj.center()` or `pobj.standardize()` will center or standardize the value axis of the record. There are also methods related to proxy system modeling: the `get_clim()` method helps to get the grid point value from a gridded climate field nearest to the record, after which the `get_pseudo()` method can be called to generate the pseudoproxy estimate. To support intuitive conversions between data objects, multiple `ProxyRecord` objects can be added together with the "+" operator to form a `ProxyDatabase` that we introduce in the next section (i.e., `pdb = pobj1 + pobj2 + ...`).

**Table 1.** The essential attributes and methods of `cfr.ProxyRecord`.

| Attributes | Description | Methods | Description |
|---|---|---|---|
| pid | The unique proxy ID. | plot() | Plot the record time series. |
| time | The time axis. | slice() | Slice the record based on a give timespan. |
| value | The value axis. | + | Combine multiple `ProxyRecord`'s as a `ProxyDatabase`. |
| ptype | The proxy type. | annualize() | Annualize/seasonalize the proxy record. |
| lat, lon, elev | The latitude, longitude, and elevation. | get_clim() | Get climate data nearest to the record. |
| seasonality | The seasonality. | get_pseudo() | Generate the pseudoproxy estimate. |
| dt | The median of time differences. | center() | Center the record against a reference timespan. |
| tags | A tag set for filtering purposes. | standardize() | Standardize the value axis of the record. |

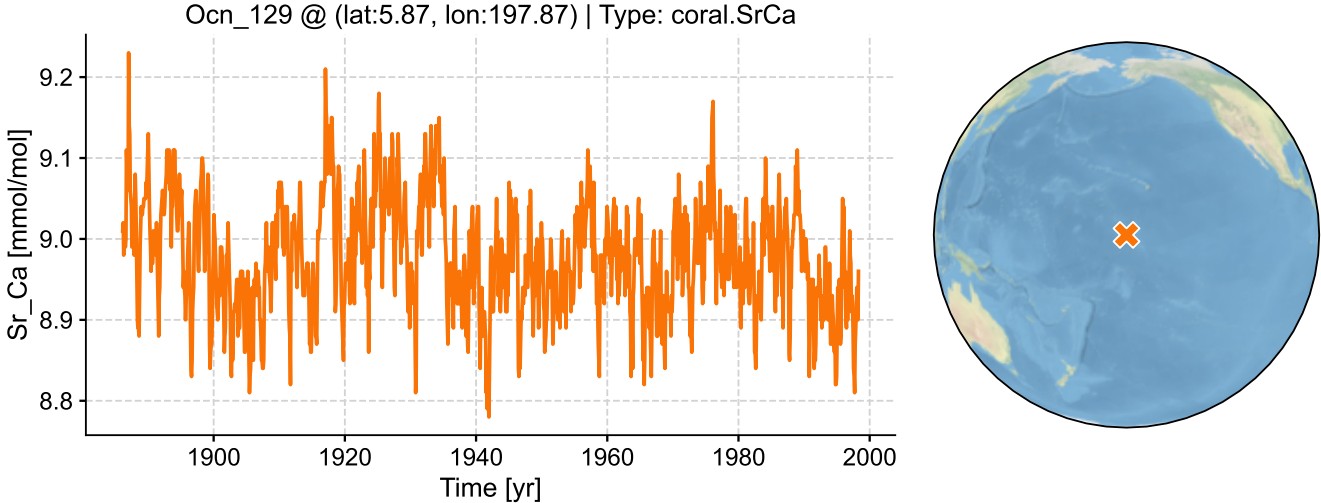

**Figure 1.** A visualization example of a coral Sr/Ca record using `cfr.ProxyRecord.plot()`.

The `ProxyDatabase` class provides a structure to organize multiple `ProxyRecord` objects at once. The essential attributes and methods are listed in Table 2. For example, suppose we have a `ProxyDatabase` object named pdb, then calling `pdb.plot()` will visualize the proxy database on a static map (Fig. 2 top), along with the count of the records by proxy type if the argument `plot_count=True` is set (Fig. 2 middle). In contrast, calling `pdb.plotly()` will visualize the proxy database on an interactive map that allows query of the location and proxy type of each record by hovering a mouse pointer (Fig. 2 bottom). The `from_df()` and `to_df()` methods allow data input and output in the form of a `pandas.DataFrame`, which is a more common format for tabular data. The `filter()` method offers a handy tool to filter the proxy database in flexible ways, such as by proxy types, latitude/longitude ranges, center location and radius distance, and arbitrary tag combinations. For more details, please refer to the notebook tutorial on this topic: https://fzhu2e.github.io/cfr/notebooks/pp2k-pdb-filter.html (Zhu et al., 2024). To

support intuitive conversions between data objects, multiple `ProxyRecord` objects and/or `ProxyDatabase` objects can be added
together with the "+" operator to form a `ProxyDatabase` (i.e., `pdb = pdb1 + pobj1 + ...`), and multiple `ProxyRecord`
objects can be removed from a `ProxyDatabase` with the "−" operator (i.e., `pdb = pdb1 - pobj1 - ...`). `ProxyDatabase`
also comes with `find_duplicates()` and `squeeze_dups()` to locate and remove duplicated records when adding multiple
databases together. In addition, with the `fetch` method, the commonly used proxy databases can be conveniently fetched
from the cloud. For instance, `pdb = cfr.ProxyDatabase.fetch('PAGES2kv2')` will remotely load the PAGES 2k phase 2
database (PAGES 2k Consortium, 2017), and `pdb = cfr.ProxyDatabase.fetch('pseudoPAGES2k/ppwn_SNRinf_rta')`
will fetch and load the base version of the "pseudoPAGES2k" database (Zhu et al., 2023a, b). Its argument can also be an
arbitrary URL to a supported file stored in the cloud to support flexible loading of the data. If the `fetch` method is called
without an argument, the supported predefined entries will be printed out to give a hint to the users.

**Table 2.** The essential attributes and methods of `cfr.ProxyDatabase`.

| Attributes | Description | Methods | Description |
|---|---|---|---|
| records | The dictionary of the records. | plot() | Plot the database on a static map. |
| pids | The list of proxy IDs. | plotly() | Plot the proxy database on an interactive map. |
| nrec | The number of the proxy records. | from_df() | Load the database from a `pandas.DataFrame`. |
| type_dict | The count of each proxy type. | to_df() | Convert the database to a `pandas.DataFrame`. |
| lats, lons | Location information in lists. | filter() | Filter the database by supported conditions. |
| | | + | Combine multiple `ProxyRecords`'s/`ProxyDatabase`'s together. |
| | | − | Removing multiple `ProxyRecord`'s from a `ProxyDatabase`. |
| | | find_dupilicates() | Find duplicated records. |
| | | squeeze_dups() | Remove duplicated records and keep one. |
| | | fetch() | Fetch and load the database from cloud. |

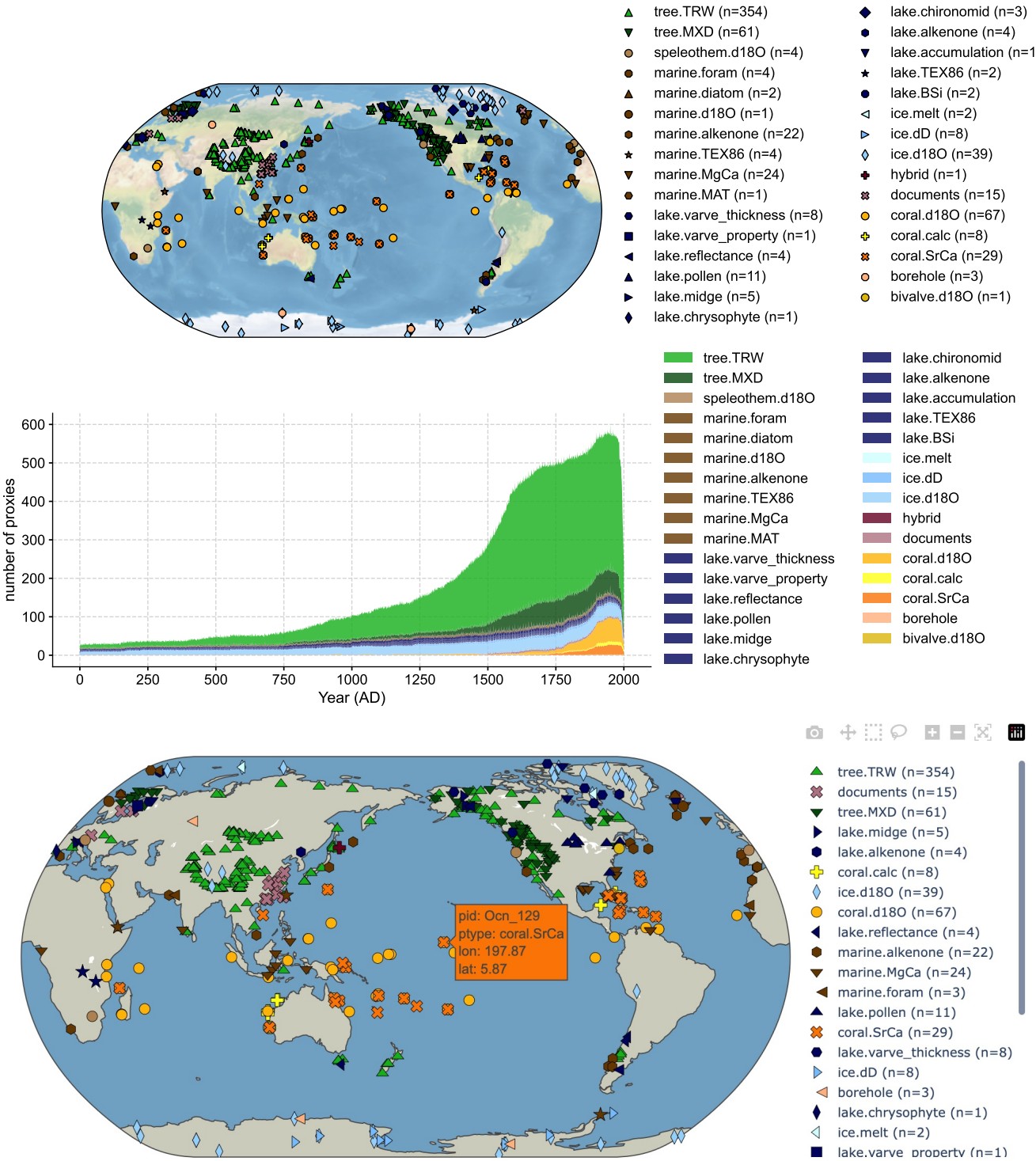

**Figure 2.** Visualization examples of the PAGES 2k version 2 multiproxy database (PAGES 2k Consortium, 2017) using `cfr.ProxyDatabase.plot()` and `cfr.ProxyDatabase.plotly()`.

## 4.2 `climate`: gridded climate data processing and visualization

The `climate` module comes with a `ClimateField` class that is essentially an extension of `xarray.DataArray` to better fit CFR applications. It contains a collection of methods for processing and visualizing the gridded climate model simulation and instrumental observation data.

The essential attributes and methods are listed in Table 3. For instance, suppose we have a `ClimateField` object named `fd`, then calling `fd.get_anom()` will return the anomaly field against a reference time period, and calling `fd.get_anom().plot()`
(an example of method cascading) will plot the anomaly field on a map (Fig. 3). The spatial resolution and coverage of the field can be altered by calling `fd.regrid()` and `fd.crop()`. Since `ClimateField` is based on `xarray.DataArray`, the slice method applies similarly; however, we augmented the original method to more robustly account for time and calendars in the paleoclimate context. For more details on time slicing, please refer to the notebook tutorial: https://fzhu2e.github.io/cfr/notebooks/climate-ops.html#Time-slicing (Zhu et al., 2024). Similar to a `ProxyRecord`, a `ClimateField` with monthly time
resolution can be annualized/seasonalized by calling the `annualize()` method with a specified list of months. To support convenient calculation of many popular climate indices such as global mean surface temperature (GMST) and NINO3.4, the `geo_mean()` method calculates the "cos-lat" weighted geospatial mean over a region defined by the latitude and longitude range. To support quick validation of a reconstructed field, the `compare()` method facilitates the comparison against a reference field in terms of a metric such as the gridpoint-wise linear correlation ($r$), coefficient of determination ($R^2$), or coefficient of
efficiency ($CE$). Lastly, the `load_nc()` and `to_nc()` methods allow loading and writing gridded datasets in the netCDF format (Rew and Davis, 1990), while the `fetch` method can be used to conveniently load gridded datasets from the cloud. For instance, `fd = cfr.ClimateField.fetch('iCESM_past1000historical/tas')` will fetch and load the iCESM simulated surface temperature field since 850 CE (Brady et al., 2019). Similar to that of `ProxyDatabase`, the argument can also be an arbitrary URL to a netCDF file hosted in the cloud, and calling without an argument will print out supported predefined entries.

**Table 3.** The essential attributes and methods of `cfr.ClimateField`.

| Attributes | Description | Methods | Description |
|---|---|---|---|
| da | The data in `xarray.DataArray`. | plot() | Plot the climate field on a map. |
| | | get_anom() | Calculate the anomaly field against a reference time period. |
| | | regrid() | Regrid the field. |
| | | crop() | Crop the field. |
| | | annualize() | Annualize/seasonalize the field. |
| | | geo_mean() | Calculate the geospatial mean over a region. |
| | | compare() | Compare against a reference field. |
| | | load_nc() | Load the field from a netCDF file. |
| | | to_nc() | Output the field to a netCDF file. |
| | | fetch() | Fetch and load the gridded climate field from cloud. |

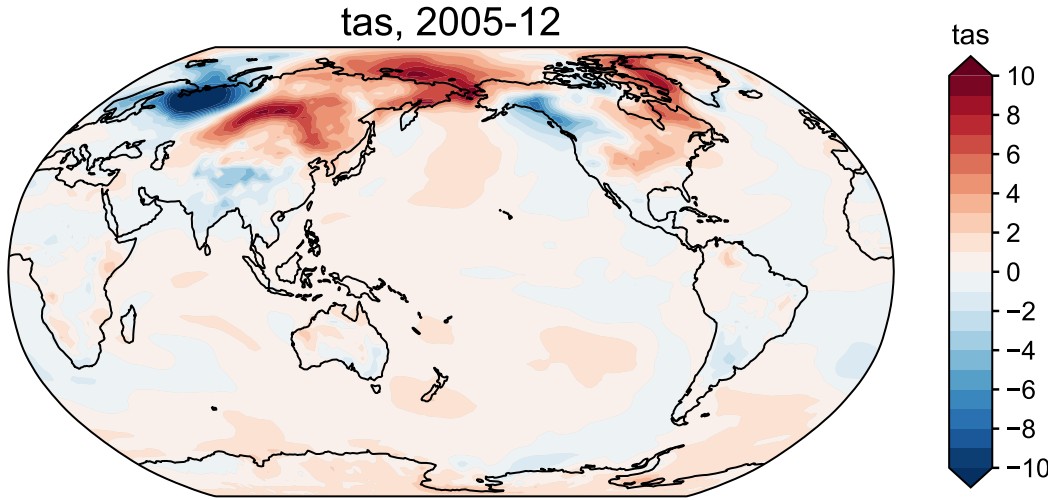

**Figure 3.** A visualization example of an iCESM simulated surface temperature anomaly field using `cfr.ClimateField.plot()`.

### 4.3  `ts`: time series processing and visualization

The `ts` module is for time series processing and visualization in general, and it comes with an `EnsTS` class to handle ensemble time series. In specific, the `EnsTS` class is mainly designed to visualize and validate the reconstructed time series such as GMST and NINO3.4, and its essential attributes and methods are listed in Table 4. The `plot()` method visualizes each member of the ensemble time series, while the `plot_qs()` method plots only the quantile envolopes (Fig. 4). Similar to a `ClimateField`, the `compare()` method is useful for a quick comparison of the ensemble median against a reference time series. The `fetch` method is supported as well. For instance, `bc09 = cfr.ProxyDatabase.fetch('BC09_NINO34')` will remotely load the NINO3.4 estimate by Bunge and Clarke (2009).

**Table 4.** The essential attributes and methods of `cfr.EnsTS`.

| Attributes | Description | Methods | Description |
|---|---|---|---|
| time | The time axis. | plot() | Plot each of the ensemble time series. |
| value | The ensemble value in matrix. | plot_qs() | Plot the quantile envolopes. |
| nt | The temporal length. | compare() | Compare against a reference time series. |
| nEns | The ensemble size. | fetch() | Fetch and load the time series data from cloud. |

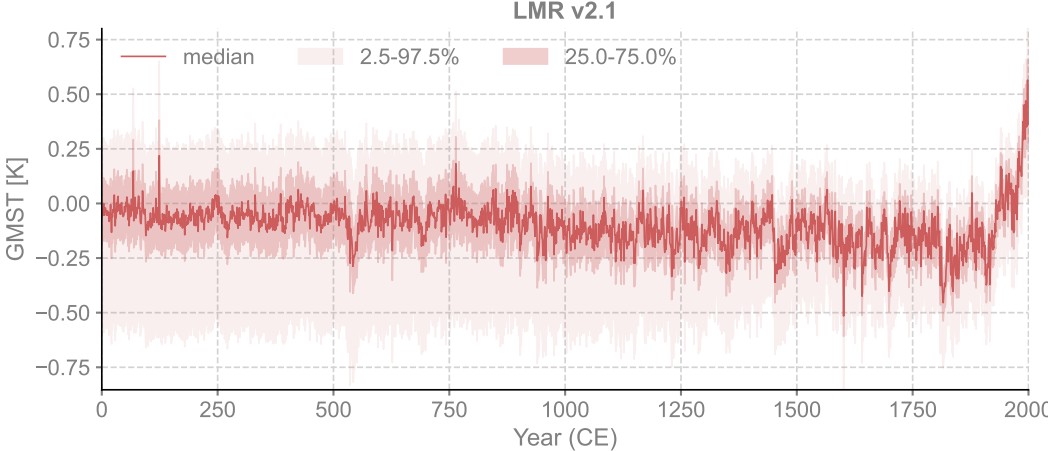

**Figure 4.** A visualization example of the LMR v2.1 ensemble global mean surface temperature (GMST) (Tardif et al., 2019) using `cfr.EnsTS.plot_qs()`.

### 4.4 `psm`: proxy system modeling

The `psm` module incorporates classes for multiple popular proxy system models (PSMs, Evans et al., 2013), including a uni-
235 variate linear regression model (Hakim et al., 2016; Tardif et al., 2019) (`Linear`), a bivariate linear regression model (Hakim et al., 2016; Tardif et al., 2019) (`Bilinear`), a tree-ring width model VS-Lite (Tolwinski-Ward et al., 2011, 2013; Zhu and Tolwinski-Ward, 2023) (`VSLite`), a simple lake varve thickness model (Zhu et al., 2023a) (`Lake_VarveThickness`), as well as the ice core $\delta^{18}$O model (`Ice_d18O`), coral $\delta^{18}$O model (`Coral_d18O`), coral Sr/Ca ratio model (`Coral_SrCa`) adopted from PRYSM (Dee et al., 2015), among which `Linear` and `Bilinear` are commonly used in paleoclimate data assimilation (PDA),
and others are more useful to generate pseudoproxy emulations (Zhu et al., 2023a, b). A summary of available PSMs in `cfr` is listed in Table 5.

Taking the default univariate linear regression model (`Linear`) as a typical example, the essential attributes and methods are listed in Table 6. As illustrated in the notebook tutorial https://fzhu2e.github.io/cfr/notebooks/psm-linear.html (Zhu et al., 2024), a `Linear` PSM object can be initialized with a `ProxyRecord` object, and by calling the `calibrate()` method, the
245 PSM utilizes the proxy measurement and the nearest instrumental observation to calibrate the regression coefficients over the instrumental period, with the option to test multiple seasonality candidates, yielding an optimal regression model. Then by calling the `forward()` method, the calibrated model will forward the nearest model simulated climate and generate the pseudoproxy estimate, translating the climate signal to the proxy space. For more illustrations of the other available PSMs, please refer to https://fzhu2e.github.io/cfr/ug-psm.html (Zhu et al., 2024).

**Table 5.** A summary of the available proxy system models (PSMs) in `cfr`.

| PSM name | Class | References |
|----------|-------|------------|
| Univariate Linear | `cfr.psm.Linear` | Hakim et al. (2016); Tardif et al. (2019) |
| Multivariate Linear | `cfr.psm.BiLinear` | Hakim et al. (2016); Tardif et al. (2019) |
| VS-Lite | `cfr.psm.VSLite` | Tolwinski-Ward et al. (2011, 2013) |
| Lake Varve Thickness | `cfr.psm.Lake_VarveThickness` | Zhu et al. (2023a) |
| Ice Core $\delta^{18}$O | `cfr.psm.Ice_d18O` | Dee et al. (2015) |
| Coral $\delta^{18}$O | `cfr.psm.Coral_d18O` | Dee et al. (2015) |
| Coral Sr/Ca | `cfr.psm.Coral_SrCa` | Dee et al. (2015) |

**Table 6.** The essential attributes and methods of the `cfr.psm.Linear` proxy system model (PSM).

| Attributes | Description | Methods | Description |
|------------|-------------|---------|-------------|
| pobj | The `ProxyRecord` object. | calibrate() | Calibrate the PSM. |
| climate_required | The list of variable names of the required climate input. | forward() | Forward the calibrated model. |

## 4.5   da: data assimilation

The `da` module implements the data assimilation algorithms, currently including the ensemble Kalman filter (Kalman, 1960; Evensen, 2009) used in the last millenium reanalysis (LMR, Hakim et al., 2016; Tardif et al., 2019) framework. The class is named `EnKF` and Table 7 lists its essential attributes and methods. Direct operations on this class are not recommended unless for debugging purposes. It is designed to be called internally by the `ReconJob` class that we introduce later for a high-level control of the workflow.

**Table 7.** The essential attributes and methods of `cfr.da.EnKF`.

| Attributes | Description | Methods | Description |
|------------|-------------|---------|-------------|
| prior | The dictionary of prior fields. | gen_prior_samples() | Generate prior samples. |
| pdb_assim | The assimilated `ProxyDatabase`. | gen_Ye() | Generate the forward estimates. |
| seed | The seed for randomization. | gen_Xb() | Generate the background matrix. |
| nens | The ensemble size. | update_yr() | Update a specific year utilizing the EnKF solver. |
| recon_vars | The names of the reconstructed variables. | run() | Run the EnKF solver through multiple years. |

### 4.6 `graphem`: Graphical Expectation Maximization

The `graphem` module implements the GraphEM algorithm (Guillot et al., 2015). The class is named `GraphEM` and Table 8 lists its essential attributes and methods. Direct operations on this class are not recommended unless for debugging purposes. Similar to `da`, `graphem` is also designed to be called internally by the `ReconJob` class.

**Table 8.** The essential attributes and methods of `cfr.graphem.GraphEM`.

| Attributes | Description | Methods | Description |
|---|---|---|---|
| field_r | The reconstructed field. | fit() | Estimate the parameters and reconstruct the climate field. |
| proxy_r | The reconstructed proxy matrix. | | |
| calib | Indices of the calibration period. | | |

### 4.7 `reconjob`: reconstruction workflow management

The `reconjob` module provides pre-defined reconstruction workflows attached to the class `ReconJob`, as listed in Table 9. We will illustrate the usage of these methods in detail in the sections on PDA and GraphEM workflows.

**Table 9.** The essential attributes and methods of `cfr.ReconJob`.

| Attributes | Description | Methods | Description |
|---|---|---|---|
| configs | The dictionary of configurations. | load_proxydb() | Load the proxy database. |
| proxydb | The loaded `ProxyDatabase`. | filter_proxydb() | Filter the proxy database. |
| prior | The dictionary of prior fields. | annualize_proxydb() | Annualize the proxy database. |
| obs | The dictionary of instrumental observations. | split_proxydb() | Split the proxy database. |
| | | load_clim() | Load the simulated or observed climate data. |
| | | annualize_clim() | Annualize the climate data. |
| | | regrid_clim() | Regrid the climate data. |
| | | crop_clim() | Crop the climate data. |
| | | calib_psms() | Calibrate the PSMs for each proxy record. |
| | | forward_psms() | Forward the PSMs for each proxy record. |
| | | run_da() | Run the DA solver. |
| | | run_da_mc() | Run DA with Monte-Carlo iterations. |
| | | prep_graphem() | Prepare data for the GraphEM solver. |
| | | run_graphem() | Run the GraphEM solver. |

## 4.8 `reconres`: result analysis and visualization

The `reconres` module focuses on postprocessing and visualization of the reconstruction results. It contains a `ReconRes` class that helps to load the reconstruction outputs stored in netCDF files from a given directory path, and organize the data in the form of either `ClimateField` or `EnsTS`. The essential attributes and methods of `ReconRes` are listed in Table 10. We will illustrate the usage in detail in the sections on PDA and GraphEM workflows.

**Table 10.** The essential attributes and methods of `cfr.ReconRes`.

| Attributes | Description | Methods | Description |
|---|---|---|---|
| paths | The paths of the reconstruction netCDF files. | load() | Load the reconstruction files. |
| recons | The dictionary of reconstruction objects. | valid() | Perform validation against given targets. |
| da | The dictionary of reconstructions in `xarray.DataArray`. | plot_valid() | Visualize the validation. |

## 5 `cfr`'s PDA workflow

In this section, we illustrate the `cfr` workflow (Fig. 5) with a reconstruction experiment taking the last millennium reanalysis (LMR, Hakim et al., 2016; Tardif et al., 2019) paleoclimate data assimilation (PDA) approach. A similar pseudoproxy reconstruction experiment can be accessed at https://fzhu2e.github.io/cfr/notebooks/pp2k-ppe-pda.html (Zhu et al., 2024).

The task here is to assimilate tropical coral records and reconstruct the boreal winter (December-February, DJF) surface temperature field, which can be used to calculate the El Niño/Southern Oscillation (ENSO) indices (NINO3.4 in this example). The "iCESM1" last millennium simulation (Brady et al., 2019) is utilized as the model prior, and the coral records from the PAGES 2k Phase 2 database (PAGES 2k Consortium, 2017) are used as the observations to update the prior. NASA Goddard's Global Surface Temperature Analysis (GISTEMP) (Lenssen et al., 2019) combining land surface air temperatures primarily from the GHCN-M version 4 (Menne et al., 2018) with the ERSSTv5 sea surface temperature analysis (Huang et al., 2017) is used as the calibration target for the PSMs. Finally, a spatially completed version of the near-surface air temperature and sea-surface temperature analyses product HadCRUT4.6 (Morice et al., 2012) leveraging the GraphEM algorithm (Guillot et al., 2015; Vaccaro et al., 2021) and the "BC09" NINO3.4 reanalysis (Bunge and Clarke, 2009) are used as the validation target for the posterior, i.e., the reconstruction.

# CFR/PDA Workflow

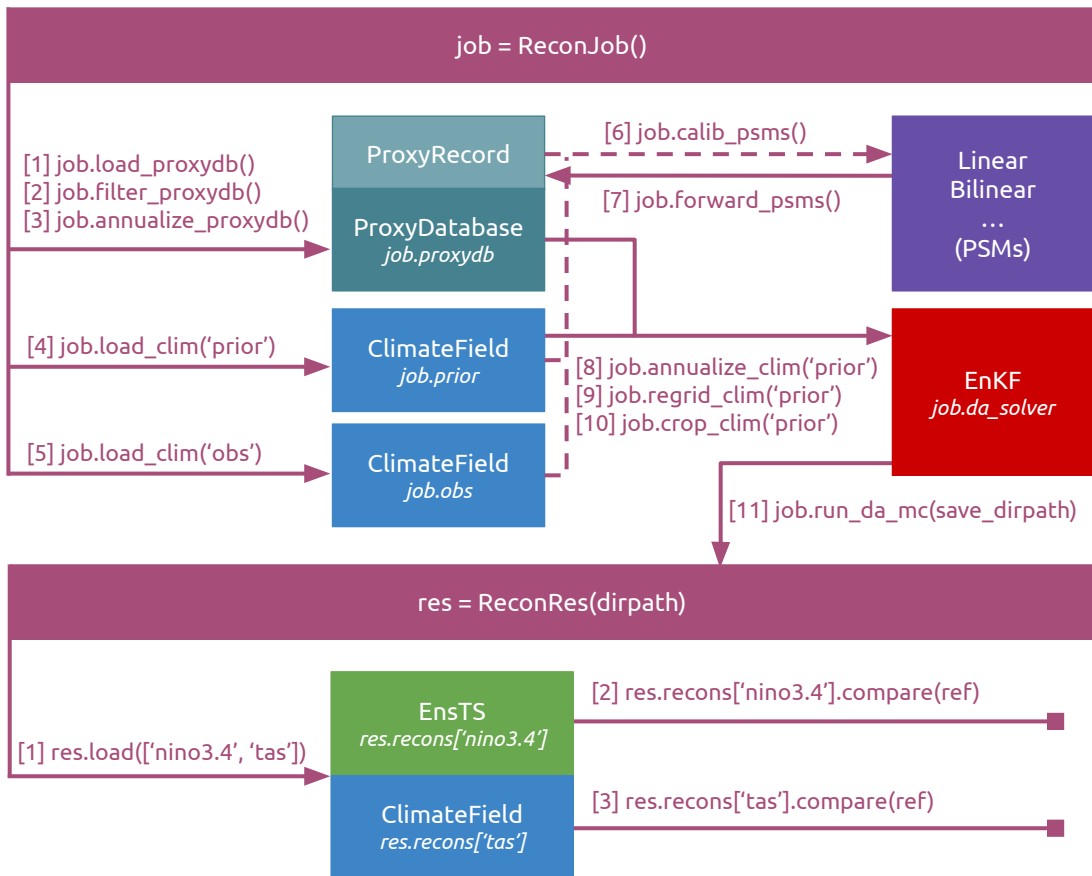

**Figure 5.** The cfr workflow for the last millennium reanalysis (LMR, Hakim et al., 2016; Tardif et al., 2019) paleoclimate data assimilation (PDA) framework. Here "nino3.4" and "tas" stand for two examples of the reconstructed variables for validation, referring to the El Niño/Southern Oscillation index series calculated as the spatial average of surface temperature within the tropical Pacific region (5°N-5°S, 170°W-120°W), and the surface temperature field, respectively. PSMs: proxy system models; EnKF: ensemble Kalman filter.

## 5.1 Load the proxy records, model simulations, and instrumental observations.

First of all, we import the `cfr` package and create a `ReconJob` object named `job`.

```
import cfr
job = cfr.ReconJob()
```

Next, we load the PAGES 2k proxy database (PAGES 2k Consortium, 2017) from the cloud and pick coral records by calling the `filter_proxydb()` method, then annualize them to be boreal winter average by calling `annualize_proxydb()`:

```
job.load_proxydb('PAGES2kv2')
job.filter_proxydb(by='ptype', keys=['coral'])
job.annualize_proxydb(months=[12, 1, 2], ptypes=['coral'])
```

Now we load the model simulated prior "iCESM1" (Brady et al., 2019) and the instrumentally observed climate data GIS-TEMP (Lenssen et al., 2019) by calling the `load_clim()` method.

```
job.load_clim(tag='prior', path_dict={'tas': 'iCESM_past1000historical/tas'}, anom_period=(1951, 1980))
job.load_clim(tag='obs', path_dict={'tas': 'gistemp1200_GHCNv4_ERSSTv5'}, anom_period=(1951, 1980),
    rename_dict={'tas': 'tempanomaly'})
```

Here, the `tag` argument is used to specify whether the loaded climate data will be used as prior or observation. The `rename_dict` argument is used to map variable names if the netCDF file does not name a variable as we assumed internally in `cfr`. For instance, we assume "lat" for latitude, "lon" for longitude, and "time" for the temporal dimension. The `anom_period` argument specifies against which time period we will calculate the anomaly.

Note that here we assume both the prior and observation fields being regularly gridded (lat/lon) datasets, which ensures a fast nearest-neighbor search in later processing steps. Any irregularly gridded datasets should be regridded beforehand using the `cfr.utils.regrid_field_curv_rect()` function or other third-party tools such as `scipy.interpolate.griddata` and ESMPy (https://earthsystemmodeling.org/esmpy).

## 5.2 Calibrate and forward the PSMs.

With the loaded proxy data, climate simulation, and instrumental observation, we are ready to calibrate the proxy system models for each proxy record by calling the `calib_psms()` method. For coral records, we use the univariate linear regression model `Linear` as defined by the `ptype_psm_dict` argument. The `ptype_season_dict`, `calib_period`, and `ptype_clim_dict` arguments specify the season, timespan (in year), and climate variables involved for calibration, respectively.

```
job.calib_psms(
    ptype_psm_dict={'coral.d18O': 'Linear', 'coral.calc': 'Linear', 'coral.SrCa': 'Linear'},
    ptype_season_dict={'coral.d18O': [12, 1, 2], 'coral.calc': [12, 1, 2], 'coral.SrCa': [12, 1, 2]},
    ptype_clim_dict={'coral.d18O': ['tas'], 'coral.calc': ['tas'], 'coral.SrCa': ['tas']},
    calib_period=(1850, 2015))
```

In this example, we have specified a fixed seasonality for the whole proxy database. Users may also perform calibration of each proxy record through a `for` loop of the proxy database, specifying various seasonality manually or based on the metadata if available. For records whose corresponding PSM is successfully calibrated, a "calibrated" tag is added to the record to facilitate filtering later. Proxy records with low calibration skill will be assigned a small weight during the assimilation process as per the EnKF algorithm, so there is no need to set a calibration threshold to manually filter the records. However, with the flexibility of the package, users have the freedom to easily filter the proxy database, using the `cfr.ProxyDatabase.filter()` method, based on any calibration threshold they like.

Once the PSMs are calibrated (and filtered), we can launch (i.e., forward) them by simply calling the `forward_psms()` method.

```
job.forward_psms()
```

## 5.3 Define the season, resolution, and domain of the reconstruction.

Now we annualize the prior by calling the `annualize_clim()` method, and specify on which resolution and domain we would like to conduct the reconstruction by calling the `regrid_clim()` and `crop_clim()` method.

```
job.annualize_clim(tag='prior', months=[12, 1, 2])
job.regrid_clim(tag='prior', nlat=42, nlon=63)
job.crop_clim(tag='prior', lat_min=-35, lat_max=35)
```

Here we annualize the prior to boreal winter, and regrid the field to a global grid with 42 latitudes and 63 longitudes, and then crop the domain to be within the 35°S to 35°N band where the corals are located.

## 5.4 Conduct the Monte-Carlo iterations of the data assimilation steps.

Once the pre-processing is complete, we are ready to conduct the data assimilation step. We call the `run_da_mc()` method to perform Monte-Carlo iterations of the EnKF assimilation steps. The argument `save_dirpath` specifies the directory path where we store the reconstruction results, and the argument `recon_seeds` specfiy the seed for randomization for each Monte-Carlo iteration.

```
job.run_da_mc(save_dirpath='./recons/lmr-real-pages2k', recon_seeds=list(range(1, 11)))
```

## 5.5 Validate the reconstruction.

Once the Monte-Carlo iterations are done, we should have several netCDF files stored in the specified directory, and we can initiate a `ReconRes` object by specifying the directory path, and load the results by calling the `load()` method.

```
res = cfr.ReconRes('./recons/lmr-real-pages2k')
res.load(['nino3.4', 'tas'])
```

Note that the variable names listed in the `load()` method are predefined. `'nino3.4'` refers to the NINO3.4 index and the reconstructed ensemble time series will be formed as a `EnsTS` object. `'tas'` refers to the surface temperature field and the ensemble mean will be formed as a `ClimateField` object. These two objects will be organized in a dictionary named `res.recons`, and their original `xarray.DataArray` forms will be stored as `res.recons['nino3.4'].da` and `res.recons['tas'].da` for more universal purposes.

To evaluate the reconstruction skill of the surface temperature field, we load a reference target named "HadCRUT4.6_GraphEM" (Vaccaro et al., 2021), which is a spatially completed version of the near-surface air temperature and sea-surface temperature analyses product HadCRUT4.6 (Morice et al., 2012) leveraging the GraphEM algorithm (Guillot et al., 2015; Vaccaro et al., 2021), and we validate both the prior and the posterior (`res.recons['tas']`) by calling the `compare()` method over the 1874-2000 CE timespan, which returns another `ClimateField` object, and we are able to visualize it by calling its `plot()` method.

```python
target = cfr.ClimateField().fetch('HadCRUT4.6_GraphEM', vn='tas').get_anom((1951, 1980))
target = target.annualize(months=[12, 1, 2]).crop(lat_min=-35, lat_max=35)

# validate the prior
stat = 'corr'
valid_fd = job.prior['tas'].compare(target, stat=stat, timespan=(1874, 2000))
fig, ax = valid_fd.plot(
    title=f'{stat}(prior, obs), mean={valid_fd.geo_mean().value[0,0]:.2f}',
    projection='PlateCarree', latlon_range=(-32, 32, 0, 360), plot_cbar=False)

# validate the reconstruction
valid_fd = res.recons['tas'].compare(target, stat=stat, timespan=(1874, 2000))
valid_fd.plot_kwargs.update({'cbar_orientation': 'horizontal', 'cbar_pad': 0.1})

fig, ax = valid_fd.plot(
    title=f'{stat}(prior, obs), mean={valid_fd.geo_mean().value[0,0]:.2f}',
    projection='PlateCarree', latlon_range=(-32, 32, 0, 360),
    plot_proxydb=True, proxydb=job.proxydb.filter(by='tag', keys=['calibrated']), plot_proxydb_lgd=True,
    proxydb_lgd_kws={'loc': 'lower left', 'bbox_to_anchor': (1, 0)})
```

With the above code lines, we get Fig. 6 top and middle. The top shows the map of Pearson correlation between the prior and the reference target ("HadCRUT4.6_GraphEM"), while the middle shows that between the posterior (i.e., reconstruction) and the reference target, indicating that the reconstruction is working as expected, boosting the mean correlation from 0.05 to 0.34. This is one measure of the information added by the proxies.

To evaluate the reconstruction skill of the NINO3.4 index, we load a reference target named "BC09" (Bunge and Clarke, 2009), and then validate `res.recons['nino3.4']` against BC09 by calling the `compare()` method, which returns another

`EnsTS` object that can be visualized by calling its `plot_qs()` method.

```python
bc09 = cfr.EnsTS().fetch('BC09_NINO34').annualize(months=[12, 1, 2])
fig, ax = res.recons['nino3.4'].compare(bc09, timespan=(1874, 2000), ref_name='BC09').plot_qs()
ax.set_xlim(1600, 2000)
ax.set_ylabel('NINO3.4 [K]')
```

The above code lines yield Fig. 6 bottom, and it indicates the reconstruction skill over the instrumental period is remarkably high, with correlation coefficient $r = 0.80$ and the coefficient of efficiency $CE = 0.57$.

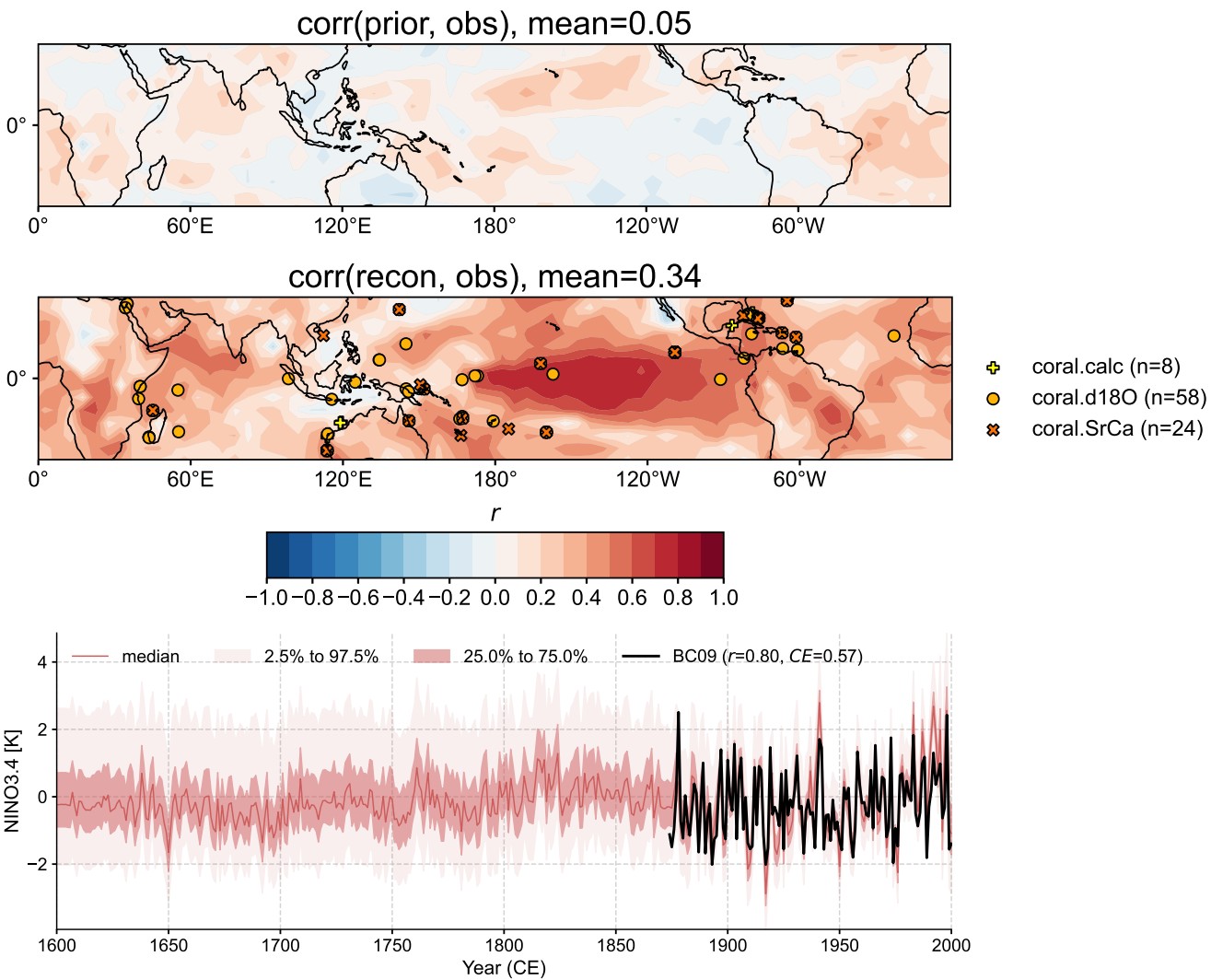

**Figure 6.** Validation of the reconstruction taking the paleoclimate data assimilation approach. (top) The correlation map between the prior filed and the observation target (Vaccaro et al., 2021). (middle) The correlation map between the reconstruction median field and the observation target. (bottom) The correlation ($r$) and coefficient of efficiency ($CE$) between the reconstructed NINO3.4 median series and the BC09 reanalysis (Bunge and Clarke, 2009). Note that the proxy calibration is performed over 1850-2015 CE, while the validation is performed over 1874-2000 CE, indicating an overlap of calibration and validation periods. The risk of overfitting is mitigated by randomly picking model prior and proxies during each Monte-Carlo iteration (Hakim et al., 2016; Tardif et al., 2019).

## 6 `cfr`'s GraphEM workflow

In this section, we illustrate the `cfr` workflow (Fig. 7) for GraphEM (Section 3.2) with a similar reconstruction experiment
setup that we presented above for the illustration of the PDA workflow. Another similar pseudoproxy reconstruction experiment
can be accessed at https://fzhu2e.github.io/cfr/notebooks/pp2k-ppe-graphem.html (Zhu et al., 2024).

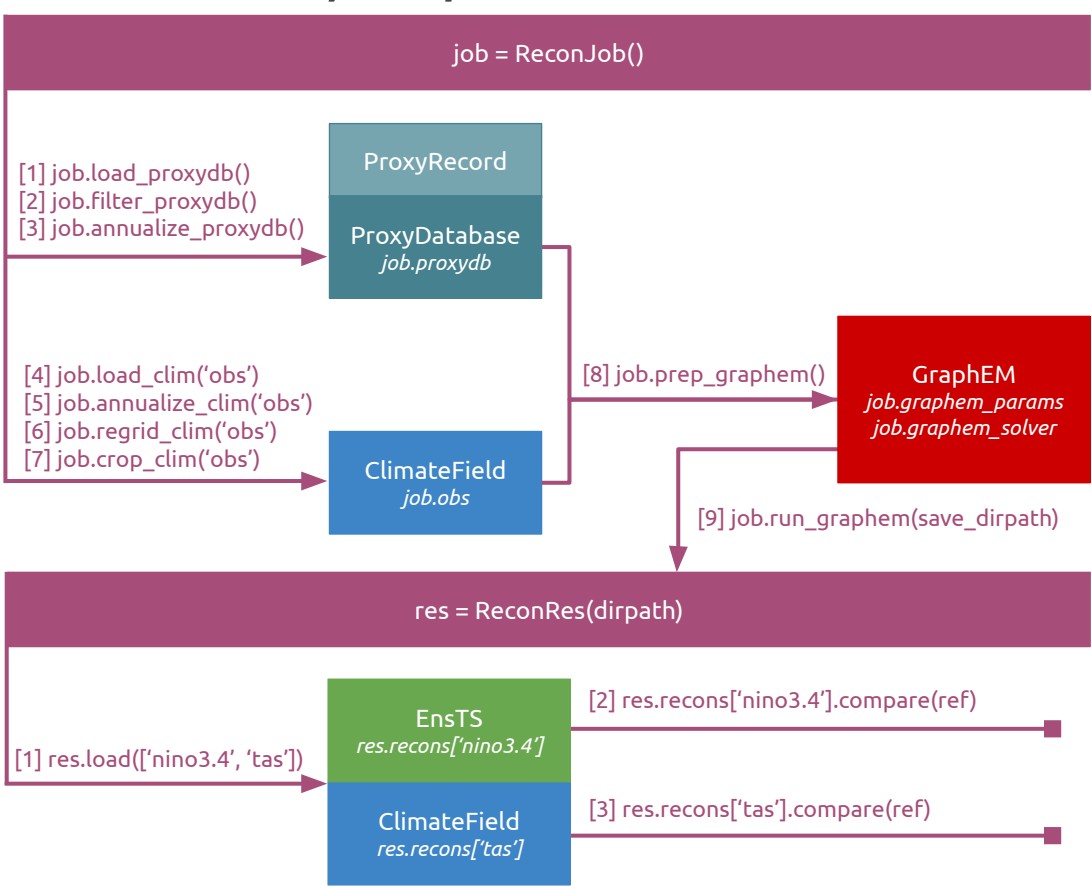

**Figure 7.** The `cfr` workflow for the Graphical Expectation-Maximization (GraphEM, Guillot et al., 2015) algorithm. Here "nino3.4" and
"tas" stand for two examples of the reconstructed variables for validation, referring to the El Niño/Southern Oscillation index series calculated
as the spatial average of the surface temperature within the tropical Pacific region (5°N-5°S, 170°W-120°W), and the surface temperature
field, respectively.

## 6.1 Load the proxy records and instrumental observations.

The `cfr` workflow for GraphEM is very similar to that for PDA. The first part is loading and preprocessing the required data, including the PAGES 2k proxy records (PAGES 2k Consortium, 2017) and the GISTEMP instrumental observation (Lenssen et al., 2019). The only difference compared to the PDA workflow is that no model prior is needed. Due to the data sparsity requirement of the GraphEM method, we set a relatively small domain around the NINO3.4 region for the reconstruction (20°S-20°N, 150°E-100°W) in this specific experiment.

```python
import cfr
job = cfr.ReconJob()

# load and preprocess the proxy database
job.load_proxydb('PAGES2kv2')
job.filter_proxydb(by='ptype', keys=['coral'])
job.annualize_proxydb(months=[12, 1, 2], ptypes=['coral'])

# load the instrumental observation dataset
job.load_clim(tag='obs', path_dict={'tas': 'gistemp1200_GHCNv4_ERSSTv5'},
    anom_period=(1951, 1980), rename_dict={'tas': 'tempanomaly'})

# define the season, resolution, and domain of the reconstruction
job.annualize_clim(tag='obs', months=[12, 1, 2])
job.regrid_clim(tag='obs', nlat=42, nlon=63)
job.crop_clim(tag='obs', lat_min=-20, lat_max=20, lon_min=150, lon_max=260)
```

## 6.2 Prepare the GraphEM solver

Compared to PDA, the preparation of the solver is much easier with GraphEM. Here, we set the calibration period to be 1901-2000 CE, while the reconstruction period is 1871-2000 CE; this reserves 1871-1900 for validation. Such dates can be easily changed with the arguments `recon_period` and `calib_period`. We also process the proxy database to be more uniform by keeping only the records that span the full reconstruction time period (`uniform_pdb=True`) to make the GraphEM algorithm more efficient.

```python
job.prep_graphem(recon_period=(1871, 2000), calib_period=(1901, 2000), uniform_pdb=True)
```

## 6.3 Run the GraphEM solver.

We run the GraphEM solver with the "hybrid" approach (Section 3.2) with a cutoff radius (`cutoff_radius`) of 5000 km for the neighborhood graph, and 2% target sparsity of the in-field part (`sp_FF`) and 2% target sparsity for the climate field/proxy

part (`sp_FP`) of the inverse covariance matrix for the GLASSO method. For more details about these parameters, please refer to Guillot et al. (2015).

```
job.run_graphem(save_dirpath='./recons/graphem-real-pages2k',
    graph_method='hybrid', cutoff_radius=5000, sp_FF=2, sp_FP=2)
```

## 6.4 Validate the reconstruction.

Similar to the PDA reconstruction experiment, we validate the reconstructed surface temperature field against the "Had-CRUT4.6_GraphEM" dataset (Vaccaro et al., 2021) and the reconstructed NINO3.4 index against the "BC09" reanalysis (Bunge and Clarke, 2009), but only over the 1874-1900 CE timespan as the reconstruction over the calibration period (1901-2000 CE) is identical to the GISTEMP observation (Fig. 8).

```
res = cfr.ReconRes('./recons/graphem-real-pages2k')
res.load(['nino3.4', 'tas'], verbose=True)

# validate the reconstructed surface temperature field
target = cfr.ClimateField().fetch('HadCRUT4.6_GraphEM', vn='tas').get_anom((1951, 1980))
target = target.annualize(months=[12, 1, 2]).crop(lat_min=-25, lat_max=25, lon_min=120, lon_max=280)

stat = 'corr'
valid_fd = res.recons['tas'].compare(target, stat=stat, timespan=(1874, 1900))
valid_fd.plot_kwargs.update({'cbar_orientation': 'horizontal', 'cbar_pad': 0.1})
fig, ax = valid_fd.plot(
    title=f'{stat}(recon, obs), mean={valid_fd.geo_mean().value[0,0]:.2f}',
    projection='PlateCarree', latlon_range=(-25, 25, 0, 360),
    plot_cbar=True, plot_proxydb=True, proxydb=job.proxydb, plot_proxydb_lgd=True,
    proxydb_lgd_kws={'loc': 'lower left', 'bbox_to_anchor': (1, 0)})

# validate the reconstructed NINO3.4
bc09 = cfr.EnsTS().fetch('BC09_NINO34').annualize(months=[12, 1, 2])
fig, ax = res.recons['nino3.4'].compare(bc09, timespan=(1874, 1900)).plot(label='recon')
ax.set_xlim(1870, 1900)
ax.set_ylim(-3, 4)
ax.set_ylabel('NINO3.4 [K]')
```

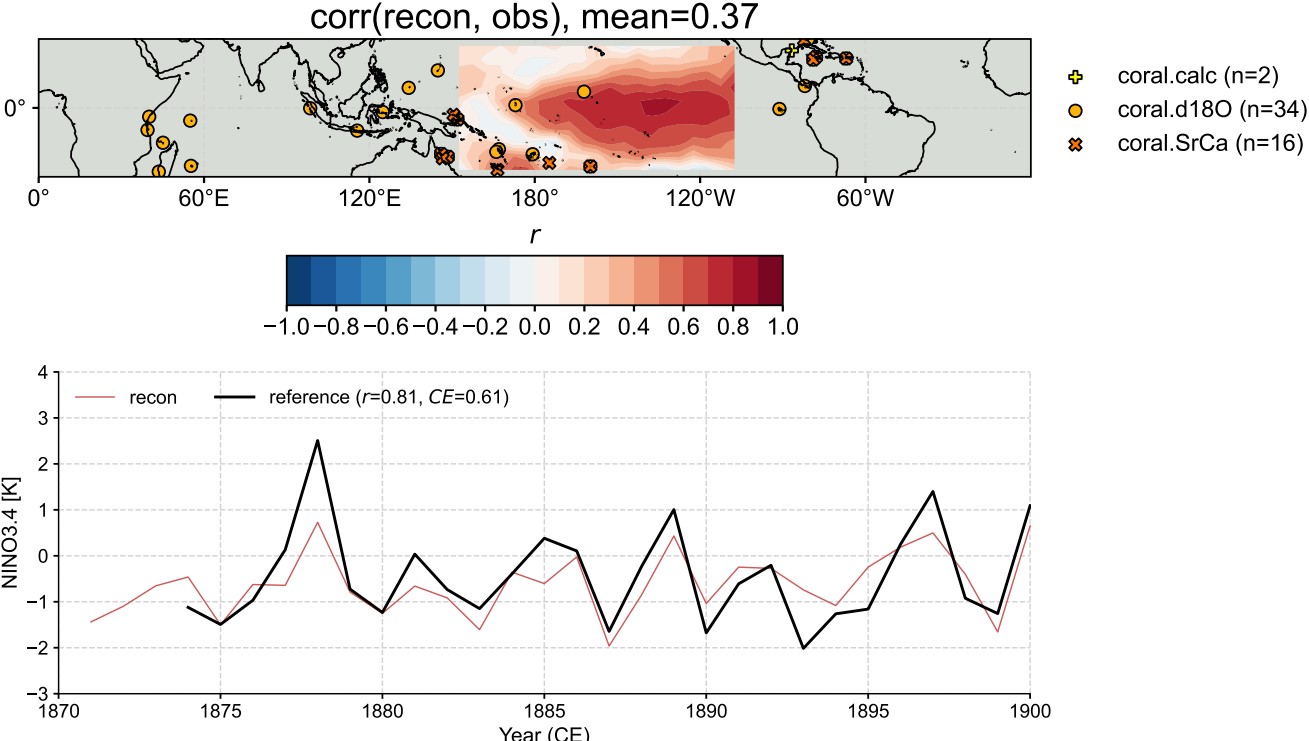

**Figure 8.** Validation of the reconstruction taking the Graphical Expectation Maximization (GraphEM) approach. (top) The correlation map between the reconstruction median field and the observation target (Vaccaro et al., 2021). (bottom) The correlation ($r$) and coefficient of efficiency ($CE$) between the reconstructed NINO3.4 median series and the BC09 reanalysis (Bunge and Clarke, 2009). Note that the calibration is performed over 1901-2000 CE, while the validation is performed over 1874-1900 CE.

The result indicates that GraphEM yields a seemingly slightly better skill compared to PDA, although we note that this
skill is sensitive to the parameters (`cutoff_radius`, `sp_FF`, and `sp_FP`), which in practice requires some tuning work in order to be optimized. In addition, we note that the current comparison is not "apples-to-apples" due to the different domains and timespans for validation. A more systematic comparison with a consistent setup is presented in the next section.

## 6.5 Comparison to PDA

A main value proposition of `cfr` is to enable convenient inter-methodological comparisons of multiple reconstructions. Here,
we illustrate how to conduct an "apples-to-apples" comparison of the reconstructions generated by the GraphEM and PDA approaches.

First, we create `ReconRes` objects for the two reconstructions and load the validation targets "HadCRUT4.6_GraphEM" (Vaccaro et al., 2021) and "BC09" (Bunge and Clarke, 2009).

```
res_graphem = cfr.ReconRes('./recons/graphem-real-pages2k')
res_lmr = cfr.ReconRes('./recons/lmr-real-pages2k')

tas_HadCRUT = cfr.ClimateField().fetch('HadCRUT4.6_GraphEM', vn='tas').get_anom((1951, 1980))
tas_HadCRUT = tas_HadCRUT.annualize(months=[12, 1, 2])
nino34_bc09 = cfr.EnsTS().fetch('BC09_NINO34').annualize(months=[12, 1, 2])
```

Then we call the `valid()` method for the reconstructions to compute the validation statistics, including Pearson correlation
and coefficient of efficiency ($CE$) against the same set of validation targets over the same domain (17°S-17°N, 153°E-108°W)
and the same time span 1874-1900 CE.

```
res_graphem.valid(target_dict={'tas': tas_HadCRUT, 'nino3.4': nino34_bc09},
    timespan=(1874, 1900), stat=['corr', 'CE'])

res_lmr.valid(target_dict={'tas': tas_HadCRUT, 'nino3.4': nino34_bc09},
    timespan=(1874, 1900), stat=['corr', 'CE'])
```

Finally, we call the `plot_valid()` method to visualize the validation results (Fig. 9).

```
fig, ax = res_graphem.plot_valid(
    target_name_dict={'tas': 'HadCRUT4.6', 'nino3.4': 'BC09'},
    recon_name_dict={'tas': 'GraphEM/tas', 'nino3.4': 'NINO3.4 [K]'},
    valid_fd_kws=dict(projection='PlateCarree', latlon_range=(-17, 17, 153, 252), plot_cbar=True),
    valid_ts_kws=dict(xlim=(1870, 1900), ylim=(-3, 4)))

fig, ax = res_lmr.plot_valid(
    target_name_dict={'tas': 'HadCRUT4.6', 'nino3.4': 'BC09'},
    recon_name_dict={'tas': 'PDA/tas', 'nino3.4': 'NINO3.4 [K]'},
    valid_fd_kws=dict(projection='PlateCarree', latlon_range=(-17, 17, 153, 252), plot_cbar=True),
    valid_ts_kws=dict(xlim=(1870, 1900), ylim=(-3, 4)))
```

It can be seen that the GraphEM and PDA approaches show overall comparable skill with a consistent spatiotemporal
validation setup, with the mean correlation skill being around 0.5 in the specific domain (Fig. 9a, d). PDA shows fairly uniform
reconstruction skill, while GraphEM's is more spatially variable, and is notably higher in the eastern equatorial Pacific. This
leads to a slightly higher reconstruction skill for NINO3.4 (Fig. 9c, f), as measured by both $r$ (0.82 vs 0.72) and $CE$ (0.63 vs
0.51).

The weak and negative correlation shown in Fig. 9a is located far from proxy locales, and could be due to two main causes,
which are not mutually exclusive:

**Covariance Modeling** The graph used here has been minimally optimized, and it is possible that it is too sparse, which would understate the strength of teleconnections. This would result in under-constrained temperature far from proxy locales, as observed here.

**Proxy Modeling** GraphEM implicitly applies a so-called "direct regression" framework (Brown, 1994), whereby the climate variable (here, `tas`) is modeled as a linear combination of all the proxies within the neighborhood identified by the graph. This stands in contrast to the "indirect regression" framework (Brown, 1994) used as part of PDA, where proxies are modeled as linearly dependent on local temperature. In the climate context, it has been shown that direct regression is problematic and may result in high variance estimates, especially when underconstrained by observations (Tingley and Li, 2012). This high variance would manifest as temperature estimates with poor correlation to the instrumental target.

Both the GraphEM and PDA based reconstructions show poor $CE$ skill over the western Pacific region (Fig. 9b, e), indicating large amplitude biases, which is likely due to issues in proxy modeling: here corals are considered temperature recorders, while in the western Pacific their signal is known to be largely driven by surface hydrology (e.g. Lough, 2010; Tierney et al., 2015).

By conducting more reconstruction experiments, we find that the PDA approach shows a robust ability to handle various spatiotemporal reconstruction setups, while the GraphEM approach is computationally limited to restricted reconstruction domains and time intervals so as to ensure a relatively dense data matrix, without which the EM algorithm can take a long time converging. This computational restriction also limits the degree of graph optimization that can be performed.

It is also worth noting that the GraphEM-based reconstruction produced by `cfr` does not, at this stage, produce ensembles. In future versions, approaches such as block bootstrapping will be implemented for uncertainty quantification, following Guillot et al. (2015).

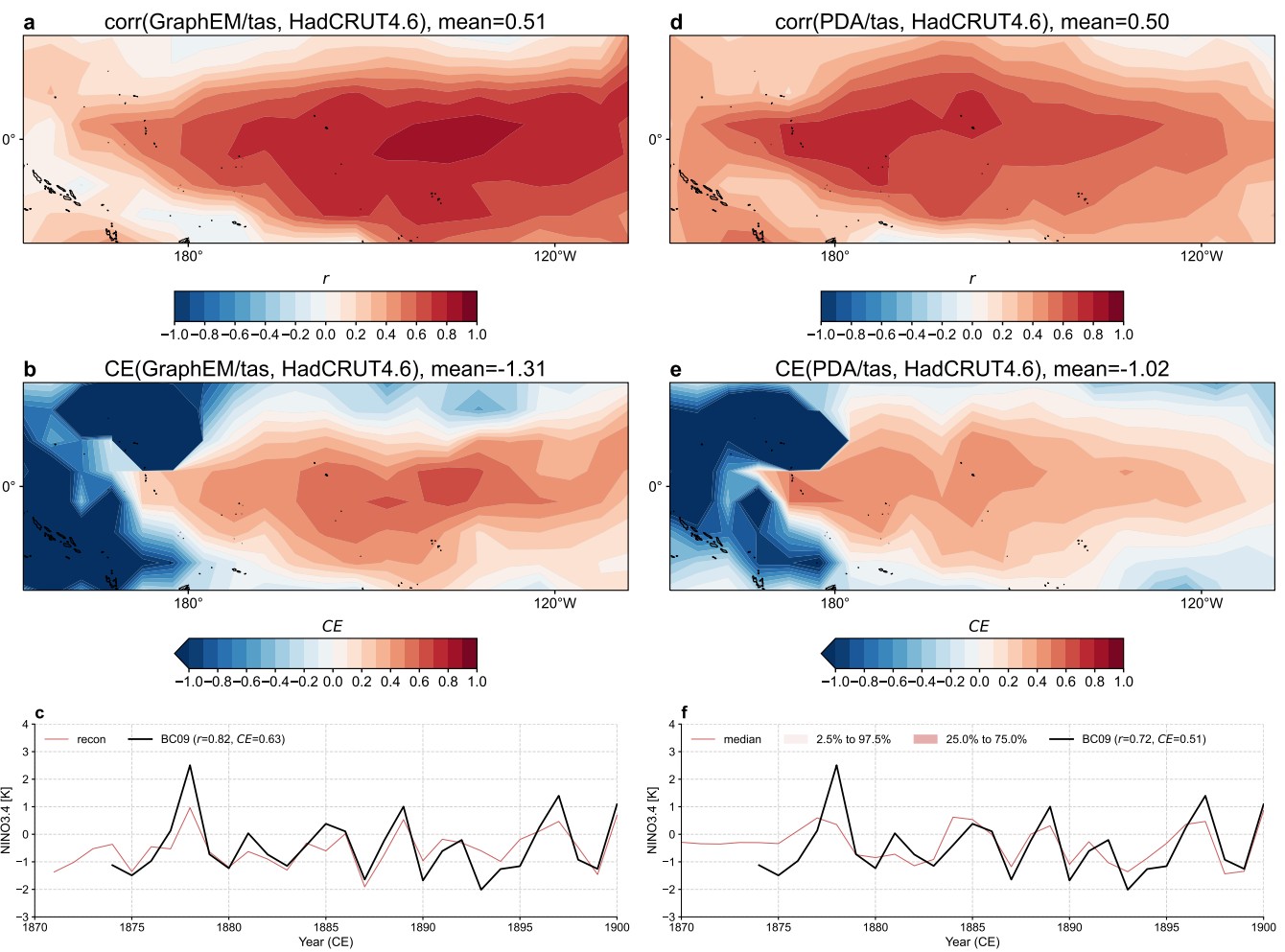

**Figure 9.** Comparison of the reconstructions generated by the GraphEM and PDA approaches against the same validation targets "Had-CRUT4.6_GraphEM" (Vaccaro et al., 2021) and "BC09" (Bunge and Clarke, 2009) over the same domain (17°S-17°N, 153°E-108°W) and the same timespan 1874-1900 CE.

## 7 Summary

Climate field reconstruction provides spatial details about past climate, and the `cfr` package makes this process intuitive, modular, and efficient for Python users. In particular, `cfr` enables a sophisticated laboratory for comparisons of different climate field reconstruction methods, which is of critical importance to better estimate and interpret reconstructions, given their sensitivity to multiple error sources rooted in the climate model prior, proxy records, proxy system modeling, and the reconstruction algorithm itself. In addition, `cfr` is a handy toolbox for paleoclimate data analysis and visualization, lowering

the bar for processing and visualizing proxy records, climate model output, and instrumental observations.

The current version of the package provides native support for climate field reconstruction over the Common Era, and its framework can be readily expanded for deep-time reconstructions (e.g., Tierney et al., 2020; Osman et al., 2021) through porting PSMs of deep-time proxies such as planktic foraminiferal $\delta^{18}O_c$, Mg/Ca, TEX'86, and UK'37 (e.g., Malevich et al., 2019; Tierney and Tingley, 2014, 2018; Tierney et al., 2019; King et al., 2023). As an open-source and community-based code, future versions of the package will combine contributions from the core development team and the community to support more CFR methods such as BARCAST (Tingley and Huybers, 2013), an online ensemble Kalman filter (Perkins and Hakim, 2017, 2021), particle filters (Goosse et al., 2010; Dubinkina et al., 2011; Dubinkina and Goosse, 2013; Liu et al., 2017; Shi et al., 2019; Lyu et al., 2021), a wider array of PSMs for deep-time reconstructions, along with more postprocessing, visualization, and validation functionalities, catalyzing open and reproducible paleoclimate research. A contributing guide is available at: https://fzhu2e.github.io/cfr. We hope this ethos invites more replicability and reproducibility in paleoclimate reconstructions, thereby deepening confidence in our knowledge of past climates.

*Code and data availability.* The `cfr` codebase is available at: https://github.com/fzhu2e/cfr under the BSD-3 Clause licence. Its documentation can be accessed at: https://fzhu2e.github.io/cfr. The exact version used to produce the results shown in this paper is archived on Zenodo at: https://zenodo.org/record/10575537 (Zhu et al., 2024), which has been tested with the current mainstream Python versions: 3.9, 3.10, and 3.11. Please follow the evolving installation guide at https://fzhu2e.github.io/cfr/ug-installation.html for more updated details, including the recommended Python version.

All datasets leveraged in the examples illustrated in this study can be automatically fetched from the cloud with `cfr`'s remote-data-loading feature, without the hassle of manually searching and downloading, including:

- The PAGES 2k phase 2 global multiproxy database (PAGES 2k Consortium, 2017) hosted on the National Center for Environmental Information's World Data Service for Paleoclimatology (https://www.ncei.noaa.gov/access/paleo-search/study/21171). A reorganized copy for the remote-data-loading purpose is hosted on Zenodo (https://zenodo.org/record/8367746) (Zhu, 2023) and Github (https://github.com/fzhu2e/cfr-data/raw/main/pages2kv2.json).

- The "pseudoPAGES2k" pseudoproxy dataset (Zhu et al., 2023a) hosted on Zenodo (https://doi.org/10.5281/zenodo.8173256) (Zhu et al., 2023b) and Github (https://github.com/fzhu2e/paper-pseudoPAGES2k).

- The "iCESM1" last millennium simulation (Brady et al., 2019) hosted on a data server at University of Washington by Rorbert Tardif (https://atmos.washington.edu/~rtardif/LMR/prior).

- The NASA Goddard's Global Surface Temperature Analysis (GISTEMP) (Lenssen et al., 2019) combining land surface air temperatures primarily from the GHCN-M version 4 (Menne et al., 2018) with the ERSSTv5 sea surface temperature analysis (Huang et al., 2017) hosted on NASA's website (https://data.giss.nasa.gov/pub/gistemp/gistemp1200_GHCNv4_ERSSTv5.nc.gz).

- The spatially completed version of the near-surface air temperature and sea-surface temperature analyses product HadCRUT4.6 (Morice et al., 2012) leveraging the GraphEM algorithm (Vaccaro et al., 2021) hosted on Zenodo (https://zenodo.org/records/4601616) and Github (https://github.com/fzhu2e/cfr-data/raw/main/HadCRUT4.6_GraphEM_median.nc).

- The "BC09" NINO3.4 reanalysis (Bunge and Clarke, 2009) hosted on Zenodo (https://zenodo.org/record/8367746) (Zhu, 2023) and Github (https://github.com/fzhu2e/cfr-data/raw/main/BC09_NINO34.csv) for the remote-data-loading purpose.

*Author contributions.* F.Z. and J.E.G. designed the research and drafted the manuscript. All authors reviewed and revised the manuscript, and contributed to the development of the package.

*Competing interests.* The authors declare no competing interests.

*Acknowledgements.* The authors acknowledge support from the National Science Foundation (grants 2303530, 2202777, and EAR 1948822 to F.Z.), and the Climate Program Office of the National Oceanographic and Atmospheric Administration (grants NA18OAR4310426 to 455 J.E.G. and NA18OAR4310422 to G.J.H.).

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
