# Peer review of "cfr (v2024.1.26): a Python package for climate field reconstruction"

_EGUsphere, 2023_

## Author Response (AR1)

**Reviews**

We are grateful to both referees for their perceptive comments, which help us improve the manuscript's quality. In the following, we address each comment in turn.

**To Referee #1**

*Main Comments:*

1. The comparison between GraphEM and PDA is insightful. I suggest providing a more detailed discussion on the similarities and differences in their results.

Response: We appreciate this suggestion. More discussion on the comparison has been made in the revision at the end of Section 6.5.

2. In Figures 6 and 9, there appears to be a weak or negative relationship between reconstructions and observations. Could you elaborate on the potential causes of this?

Response: Thank you for pointing out this issue. There are three aspects that could cause such a weak/negative relationship between reconstructions and observations:

(i) Covariance modeling. In the case of PDA, the spatial covariance structure in model prior could be incorrect compared to instrumental observations in certain regions; in the case of GraphEM, the graph being used in our reconstruction example is minimally optimized, and possibly too sparse.

(ii) Proxy modeling. In the case of PDA, we applied a simple univariate linear regression model to forward modeling the coral records, which works great for regions where the signal is mainly surface temperature, but can work poorly where the signal is largely driven by surface hydrology.

(iii) The validation target. We realize that it might be inappropriate to validate our reconstructions against the 20CRv3 surface temperature reanalysis. Since our coral records are essentially calibrated against ERSSTv5, our reconstructions over the tropical Pacific are thus SST instead of air surface temperature. While 20CRv3 is good for air surface temperature validation, it might not be the best for SST validation. Therefore, in our revision, we have replaced our validation target to be a spatially completed version of HadCRUT4.6 (Morice et al., 2012). With this more appropriate validation target, the issue pointed out by this comment is largely alleviated.

We have added the discussion of the first two bullets in our revision in L386-399.

1. Line 4: The term "time-consuming" could be replaced with "not transparent enough" to better reflect the challenges in current methods.

Response: Thank you for the suggestion. We used the word "obscure" in our revision due to its conciseness.

2. Lines 29-30: Consider expanding the discussion on Bayesian methods and inverse/indirect regression by citing Tingley and Huybers (2010a, b, 2013), and (Christiansen and Ljungqvist, 2017; Shi et al., 2017).

Response: Thank you for suggesting the additional references. We have cited them in our revision.

3. Line 82: Please include recent works on data assimilation such as(Fang et al., 2022; Lyu et al., 2021; Shi et al., 2019).

Response: The additional references have been cited in the revision.

4. Figure 5: Introducing simple physics terminology could enhance the clarity of the flow chart for a broader audience.

Response: The captions of the flowcharts (the original Fig. 5 and Fig. 7) have been augmented with physics terminology and additional explanations to cater to a wider audience.

5. Line 371: The list of references on particle filters could be updated to include (Lyu et al., 2021; Shi et al., 2019).

Response: The reference list has been updated according to the suggestion.

**To Referee #2**

1. The package and the manuscript target specific time scales, namely the past centuries or past millennium, for which annually resolved, perfectly dated proxies exist. It does not target other types of longer time-scale reconstructions, such as for the Pleistocene or even further back in time, for which only proxies with uncertain dating and varying time resolution are available. These latter reconstructions require different reconstruction methods, and the calibrations are much more difficult. Some calibrations for those types of proxies are based on empirical formulae that do not undergo a data-based calibration. This limitation should be made explicit, perhaps in the title, but certainly in the abstract

Response: This is a good point. The PDA framework utilized in this package also works for deep-time climate field reconstruction(see e.g., Tierney et al., 2020; Osman et al., 2021). The only missing part is the PSMs for deep-time proxy records such as marine sediments, which is readily available (see e.g., https://github.com/brews/baymagpy, https://github.com/brews/bayfoxr, https://github.com/brews/baysparpy). We plan to include these deep-time PSMs in the future as mentioned in the Summary section in the original manuscript. We have added more clarifications in our revision in the last paragraph of the Summary section to make this clearer.

2. It remains unclear whether the package considers the situation in which proxy records have different coverage periods within a given set of proxies, e.g. corals. This is related to whether the package considers the possibility of missing data, both in the proxy records and the instrumental data used for calibrations.

Response: Thank you for pointing out this matter. Yes, the package is designed around missing data.

The PDA framework treats each target year for reconstruction in a standalone manner, meaning the reconstruction of a certain year will not affect that of the other years. This design alleviates the issue of non-uniform temporal data availability, algorithmically speaking, and it makes parallel reconstruction of multiple years easier to conduct. In our revision, we add this clarification to the last paragraph of Section 3.1 to make this clearer.

GraphEM is also designed around infilling missing data. They are only problematic over the instrumental period, where they must be infilled using the "neighborhood graph" method prior to launching a reconstruction, as mentioned in the original last paragraph of Section 3.2.

3. A further question is the seasonality of the proxy records. The seasonality has to be the same for all proxy records within a data set. This would then require that the seasonality is prescribed at the outset of the reconstructions and that the final product is, e.g., a winter reconstruction. Alternatively, the user would go for an annual reconstruction, even if the proxies are known to have a specific seasonality. In this case, would the package select the proxies that do not pass a calibration threshold? It would, but the reader would need clarification on it.

Response: This is a great point. In the PDA framework, proxy records with low calibration skill are assigned a small weight for assimilation as per the EnKF algorithm, so there is no need to set a threshold. However, with the flexibility of the package, users also have the freedom to easily filter the proxy database based on any calibration threshold they like. In addition, the proxy database can be filtered during the processing step based on the metadata if available. Clarifications have been added to the revision in Section 5.2.

4. Python versions are sometimes a headache for the user. Which Python version is required by the package, or which is the last Python version under which the package would not run?

Response: The cfr package has been tested with the currently mainstream versions: 3.9, 3.10, and 3.11. The installation guide (https://fzhu2e.github.io/cfr/ug-installation.html) shows the recommended version (currently 3.11), and ensures an easy installation for users. Version 3.12, the current latest version, can not work properly due to the fact that it utilizes the Python scientific stack (e.g., Numpy, Scipy, Xarray, etc.) which usually takes some time to support the latest version. We have added clarification in our revision in the section for code and data availability.

5. The manuscript includes two reconstruction methods so far. Including more requires quite a bit of work. What are the future plans? Let the community develop further packages. Are the authors themselves expanding the method's base?  Given the authors ' background, an obvious next candidate could be BARCAST, and if yes, it can be useful to the community that such an additional package is in the pipeline. Alternatively, the software repository could allow other groups to pre-announce what they are working on to avoid duplicates and a waste of resources.

Response: Thank you for this great point. Yes, as mentioned in the last paragraph of the Summary section, the package is open-source and publicly hosted on Github, a platform where users may submit issues and/or solutions, propose ideas, and push code updates. We have the plan to include more methods, including BARCAST, and in the meantime, we encourage the community to contribute to its future development. In our revision, we mention that a contributing guide is available at: https://fzhu2e.github.io/cfr

6. When using the observations to calibrate the proxies, the example uses a gridded data set, in which I guess that the method selects the grid cell closest to the proxy location (or a spatial interpolation?). Is it possible to set the selection procedure for observational data sets?   Some observational data sets, e.g., high-resolution reanalysis, are produced with regional climate models with curvilinear coordinates, so this may become an issue in areas of complex topography.  Alternatively, the calibration would use irregularly located station data, so the user would prefer the nearest-neighbor selection procedure, suppressing automatic interpolation.

Response: Thank you for pointing out this issue. The current framework indeed assumes that the observations utilized for proxy calibration are stored as a regularly gridded (lat/lon) dataset because such a setting ensures a fast nearest-neighbor search. Any other irregular grid will require a more complex search algorithm and will slow down the entire workflow. Therefore, we recommend a regridding process beforehand if users have

observational data that is irregularly spaced. We have added clarification in the last paragraph of  Section 5.1 in our revision.